# Figuring Method of High Convergence Ratio for Pulsed Ion Beams Based on Frequency-Domain Parameter Control

**DOI:** 10.3390/mi13081159

**Published:** 2022-07-22

**Authors:** Lingbo Xie, Ye Tian, Feng Shi, Gang Zhou, Shuangpeng Guo, Zhe Zhu, Ci Song, Guipeng Tie

**Affiliations:** 1College of Intelligence Science and Technology, National University of Defense Technology, Changsha 410073, China; lingbotse@163.com (L.X.); shifeng@nudt.edu.cn (F.S.); zg2206553079@foxmail.com (G.Z.); g18231033287@163.com (S.G.); a362521096@163.com (Z.Z.); sunicris@163.com (C.S.); tieguipeng@163.com (G.T.); 2Hunan Key Laboratory of Ultra-Precision Machining Technology, Changsha 410073, China; 3Laboratory of Science and Technology on Integrated Logistics Support, National University of Defense Technology, Changsha 410073, China; 4Laboratory of Thin Film Optics, Shanghai Institute of Optics and Fine Mechanics, Chinese Academy of Sciences, Shanghai 201800, China

**Keywords:** pulsed ion beam, high convergence ratio, large gradient error figuring

## Abstract

The continuous phase plate (CPP) provides excellent beam smoothing and shaping impacts in the inertial confinement fusion application. However, due to the features of its dispersion, its surface gradient is frequently too large (>2 μm/cm) to process. When machining a large gradient surface with continuous ion beam figuring (IBF), the acceleration of the machine motion axis cannot fulfill the appropriate requirements, and the machining efficiency is further influenced by the unavoidable extra removal layer. The pulsed ion beam (PIB) discretizes the ion beam by incorporating frequency-domain parameters, resulting in a pulsed beam with a controlled pulse width and frequency and avoiding the extra removal layer. This research evaluates the processing convergence ability of IBF and PIB for the large gradient surface using simulation and experiment. The findings reveal that PIB offers obvious advantages under the same beam diameter. Compared with the convergence ratio (*γ* = 2.02) and residuals (RMS = 184.36 nm) of IBF, the residuals (RMS = 27.48 nm) of PIB are smaller, and the convergence ratio (*γ* = 8.47) is higher. This work demonstrates that PIB has better residual convergence in large gradient surface processing. It is expected to realize ion beam machining with a higher convergence ratio.

## 1. Introduction

The focus radiation of an inertial confinement fusion device must be highly homogenous [1,2]. The instability of fluid mechanics magnifies the uneven distribution of light intensity, resulting in a variety of nonlinear consequences. Continuous phase plate (CPP) is a phase-type diffractive optical element that can efficiently adjust the size of the focused spot and utilize more than 98 percent of the incident light by modifying the incident light’s wavefront. However, due to its undulating morphology at the micron or even nanometer level, the convergence of the discrepancy between the processed and designed shapes has become a machining challenge [3].

The processing method represented by magnetorheological finishing (MRF) technology has achieved good results in the processing of large-diameter CPP components [4]. However, the machined CPP feature morphology is constrained due to the huge polishing wheel and physical dimension constraints. It cannot handle fine structures with small feature size (<4 mm) and large local surface undulation (>2 μm/cm) [5,6,7].

Ion beam figuring (IBF) realizes the change of material removal amount in different processing regions by controlling the dwell time of the ion beam on the surface of the component. By changing the aperture of the ion beam, the accurate machining of small characteristic sizes can be achieved [8,9,10].

However, when there is a large gradient surface error, the corresponding dwell time changes greatly, and it is difficult to match the speed change in finite time only by the acceleration of the moving axis of the machine tool [11]. There is a certain error between the actual and ideal speed, and thus the material cannot be removed accurately. Therefore, although the removal resolution of this processing method can theoretically achieve nano accuracy, the convergence ratio (ratio of preprocessing error to postprocessing error) of residuals is still low in practical machining in actual processing [12].

Based on the current status of ion beam processing, we used pulse power to discretize the continuous ion beam of IBF. The controllable pulse width and frequency were realized by introducing frequency-domain parameters. The parameters that control the amount of material removed are transformed from dwell time to duty cycle. As a result, by adjusting the duty cycle in real time, the amount of material removed from each scanning grid can be precisely regulated [13].

This material removal method based on duty cycle adjustment enables the motion axis to move uniformly during the processing [14,15,16], completely avoids the restriction of the acceleration of the machine tool motion axis and reduces the requirement for the dynamic performance of the machine tool while having high machining convergence ability [17,18,19]. Previously, Zhou et al. [13] only validated the stability and linearity of the PIB removal function; its actual shaping ability and advantages were not verified.

In Section 2, we compare the modification theory of IBF and PIB and then propose a method to solve the pulse duty cycle matrix based on pulse iteration and modify the duty cycle matrix by optimizing the relaxation factor. In Section 3, we conduct an actual CPP surface figuring. The results verify the correctness of the simulation algorithm and the modifying advantage of PIB over IBF, which has guiding significance for the breakthrough of ion beam processing to achieve higher efficiency.

## 2. PIB Machining Modification Theory

The continuous ion beam figuring method subtracts the target surface from the initial surface to obtain the expected removal and then grids the data. By inverse convolution, the removal function and removal amount of each grid point, the dwell time of each grid point is determined. By changing the speed of the machine tool motion axis, the dwell time of each grid point can be adjusted to realize the change of the corresponding point processing removal [20,21,22].

In this way, the corresponding dwell time obtained by inverse convolution is zero for regions that need not be removed—that is, the speed of the machine tool axis is infinite at this time, which is impossible to achieve. Therefore, there will be an extra removal layer with this calculation method, which means that the area where the component does not need to be removed will still be sputtered by the ion beam. As a result, the efficiency of machining error convergence is affected, and the modification ability of ion beam machining is reduced.

The PIB solves this problem by changing the principle of material removal. The key parameter matrix that controls the variation of the resection is the duty cycle (DC) matrix, which can contain zero values [22]. After the duty cycle corresponding to each grid point is obtained, the plasma will be beamed according to the calculated duty cycle matrix. When the duty cycle is greater than zero, the plasma is emitted from the ion sheath, and the material is removed by sputtering on the surface of the component. The amount of removal is linearly related to the duty cycle [13]. When the duty cycle is zero, the plasma is confined in the ion sheath without material removal.

To solve the duty cycle from Figure 1d,e, we use the matrix-based pulse iterative method to calculate the duty cycle matrix. Considering the processing efficiency and solution error, the surface is meshed by a 1 mm interval, and the two-dimensional surface error can be decomposed into a splice of *n* continuous one-dimensional errors.

Assume that there is now a sinusoidal error with wavelength λ and amplitude δ, then the expected material removal in the X direction is H(x):(1)$$H\left(x\right)=\delta \left[\sin\left(\frac{2\pi x}{\lambda }\right)+1\right]$$

Assuming that the machine moves at a uniform speed v during processing, the scanning grid is subdivided into l when the dwell time is solved. The expected material removal can be achieved by convoluting the duty cycle matrix with the removal function, and Equation (1) becomes:(2)$$H\left(x\right)=T\cdot DC\left(x\right)*R\left(x\right)=\frac{l}{v}\cdot DC\left(x\right)*R\left(x\right)$$
where DC(x) is duty cycle vector, one-dimensional removal function R(x) given by:(3)$$R\left(x\right)=\frac{B}{\sqrt{2\pi \sigma }}exp(-\frac{x^{2}}{2\sigma^{2}})$$

*B* is the removal rate of the function volume, and σ is the removal function Gaussian distribution parameter. The duty cycle vector can be obtained by the deconvolution calculation:(4)$$DC\left(x\right)=\frac{v}{l}\cdot \frac{\delta }{B}\left[exp{\left(\frac{\sqrt{2}\pi \sigma }{\lambda }\right)}^{2}\sin\left(\frac{2\pi x}{\lambda }\right)+1\right]$$

Since the duty cycle cannot be negative, Equation (4) is modified to:(5)$$DC=\frac{v}{l}\cdot \frac{\delta }{B}\cdot exp{\left(\frac{\sqrt{2}\pi \sigma }{\lambda }\right)}^{2}\left[\sin\left(\frac{2\pi x}{\lambda }\right)+1\right]$$

Combining *n* non-negative corrected one-dimensional duty cycle vectors to obtain the matrix and convoluting the function matrix to obtain the total processing removal at this time, the removal amount corresponds to a non-negative duty cycle matrix, which is usually not completely consistent with the actual processing removal amount. Therefore, the relaxation factor ξ is introduced into the removal amount of simulation calculation, and the duty cycle matrix is revised again. The residual error is obtained by subtracting the target surface from the shape after processing:(6)$$E_{0}=H-DC_{0}*R,k=0$$

Set the duty cycle matrix correction $\Delta_{k}=\frac{E_{k}}{B}$, and then the duty cycle matrix is corrected by pulse iteration method:(7)$$DC_{k+1}=DC_{k}+\xi \Delta_{k}$$
(8)$$E_{k+1}=H-DC_{k+1}*R$$

The fitting residuals are given by:(9)$$E_{\xi }=E_{0}-E_{k+1}$$

If $E_{\xi }$ does not meet the requirement, let *k* = *k* + 1 continue to iterate until the fitting residual $\left|E_{\xi }\right|$ reaches the maximum. The practicability of this algorithm will be verified in Section 3.

## 3. PIB Modifying Capability Verification

In this section, we verify the convergence effect of machining residuals for actual workpieces. To increase the efficiency, we processed fused quartz samples with a diameter of 100 mm using MRF. Figure 2 depicts the matching residual after MRF as well as the design surface. The MRF processed sample is then further changed with IBF and PIB, and the matching residuals between the two modified samples and the design surface are compared.

In Figure 2b, the wavefront gradient of the matching residual is calculated as 2.2 μm/cm, which satisfies the definition of large gradient in Section 1. The design surface structure corresponds to a frequency band *f* < 0.1 mm^−1^ as shown in Figure 2c. Considering the machining efficiency, the ion beam with a 15 mm diameter is selected for processing with a cut-off frequency *f =* 0.13 mm^−1^, which fulfills the requirements of modification. We use the simulation algorithm to calculate the residuals of IBF and PIB. The final duty cycle matrix and relaxation factor (ξ=1.003) are obtained from the iterative algorithm in Section 2. The residuals are shown in Figure 3.

Usually, the RMS value of residual is used to characterize the superiority of machining. The smaller the RMS value, the less difference there is between the machined surface and the designed surface. The simulation results in Figure 3 show that PIB has distinct advantages over IBF since IBF cannot reach a dwell time of zero. As a consequence, the time taken by the machine tool to pass through the machining area at the fastest speed is used to instead of the calculated zero value, and thus the removal is still formed in areas that should not have been removed, which leads to worse results in the IBF simulation.

To verify the correctness of the simulation, we conducted an actual machining experiment. In order to improve the efficiency and make the result difference obvious, we divide the whole sample into two parts: the right part was processed by IBF, and the left part was processed by PIB. Considering the removal of the function beam diameter, the overlapping area with 10 mm in the middle was not processed to avoid unnecessary errors in the residual calculation after processing on the left and right sides. The pulse frequency should satisfy the condition that the machine tool passes through each grid in a pulse period (f<v/l). The complete machining parameters are shown in Table 1, and the residuals after processing are shown in Figure 4.

From Figure 4 and Figure 5, the residual PV and RMS values after PIB processing are smaller than those after IBF processing. The residual PV after IBF processing is 1383.92 nm, the RMS is 184.36 nm, and the convergence ratio *γ* is 2.02. Regarding the area after PIB processing, the residual PV after PIB processing is 360.75 nm, the RMS is 27.48 nm, and the convergence ratio *γ* is 8.47.

In the experiment of the actual CPP surface, PIB showed excellent processing convergence because of its unique material removal process. It allows the ion beam to be “cut off” during the process so that the beam is elicited only in areas requiring removal. The removal amount of the corresponding region can also be matched by adjusting the duty cycle. This ‘fixed-point removal’ processing method has better dynamic adaptability than IBF, and its convergence efficiency of single processing is significantly improved. Therefore, compared with 2.02 of IBF, the convergence ratio of PIB increases to 8.47, which is significantly improved.

## 4. Discussion

In this study, the material removal principle of traditional IBF is changed by introducing frequency-domain parameters. We propose a theory of pulse ion beam figuring based on duty cycle control and a duty cycle matrix calculation method, as well as a relaxation factor to modify the simulation results. There are certainly acceptable deviations between the results of actual processing and simulation. The reason may be that only half of the sample is actually processed, while the simulation residual calculation is for the entire sample surface. Under this variation, the simulation algorithm in Section 2 might be regarded for guiding the importance for actual processing.

Another advantage of PIB over IBF is that it requires fewer dynamic characteristics of the machine. In the experiments, the same beam diameter was chosen for the IBF and PIB, and the beam diameter of this size is fully machinable for the expected removal amount. However, the difference in residuals between the two different processing methods in the results was significant. The reason may be that the processing mode of IBF is based on the change of dwell time. When the processing amount changes greatly in a short time, the movement speed of the machine tool will change dramatically. If the machine tool’s motion characteristics cannot match the change, the real processing removal amount will not reach the target value as shown in Figure 6. The ion source will also fluctuate when moving rapidly, which will change the removal efficiency.

In contrast, the unique processing method of PIB enables it to become a processing method with a higher convergence ratio.

## 5. Conclusions

PIB has unique advantages in the ultra-high precision improvement of optical components. The derivation of its modification theory and the verification of its convergence ability have important guiding significance for its practical processing application. In this paper, the calculation method of the duty cycle matrix and relaxation factor were obtained by deducing the modification theory based on the duty cycle change. The excellent modification convergence ability of PIB for large gradient complex surfaces was verified.

Through the comparison of the actual CPP surface modification, the residuals also show that PIB (*γ* = 8.47) had a higher convergence ratio than IBF (*γ* = 2.02); thus, PIB is a superior processing method to achieve a higher convergence ratio of the optical components. The PIB has the potential to become the next generation of ion beam modifying tools with its “spot removal” characteristics and lower dynamic requirements of the machine tool, and this is important for the further development of ultra-precision machining.

## Figures and Tables

**Figure 1 micromachines-13-01159-f001:**
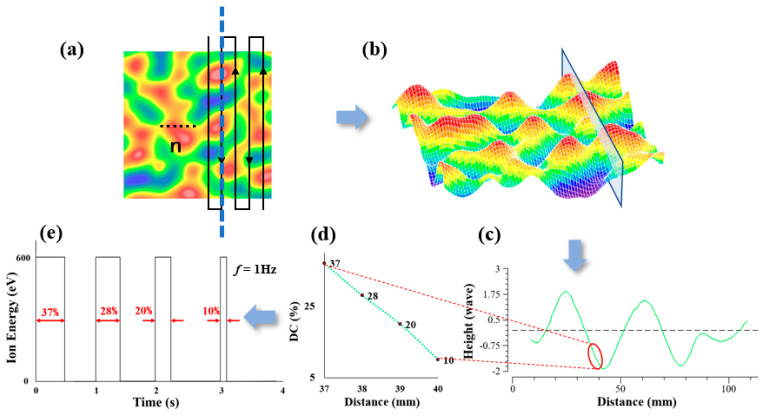
(**a**) Processing removal amount. (**b**) The section corresponding to the dotted line. (**c**) Section surface shape at dotted line. (**d**) Duty cycle corresponding to the surface in circle. (**e**) Duty cycle to voltage.

**Figure 2 micromachines-13-01159-f002:**
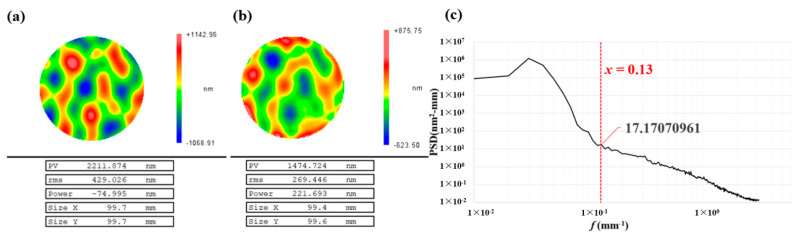
(**a**) Design surface. (**b**) Matching residual after MRF. (**c**) PSD curve of the design surface.

**Figure 3 micromachines-13-01159-f003:**
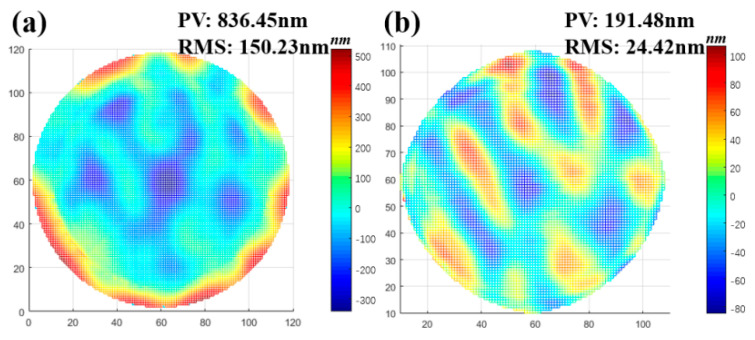
(**a**) Residuals after IBF simulation. (**b**) Residuals after PIB simulation.

**Figure 4 micromachines-13-01159-f004:**
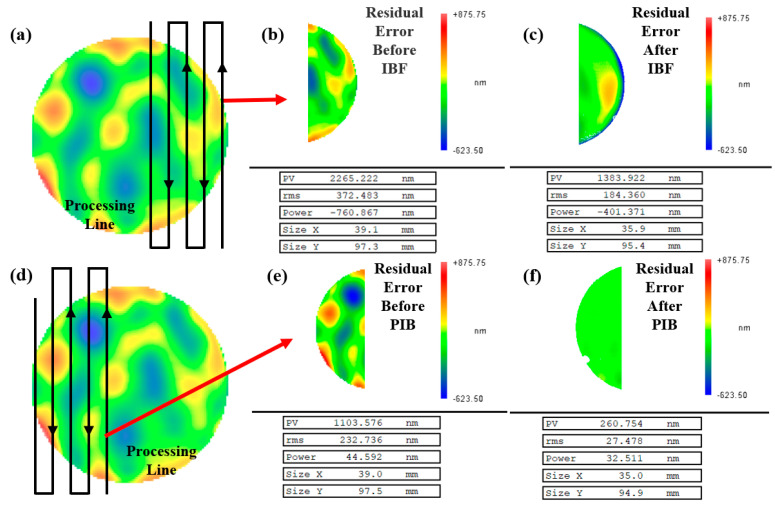
(**a**) Actual machining area of IBF. (**b**) Residual before IBF processing. (**c**) Residual after IBF processing. (**d**) Actual machining area of PIB. (**e**) Residual before IBF processing. (**f**) Residual after PIB processing.

**Figure 5 micromachines-13-01159-f005:**
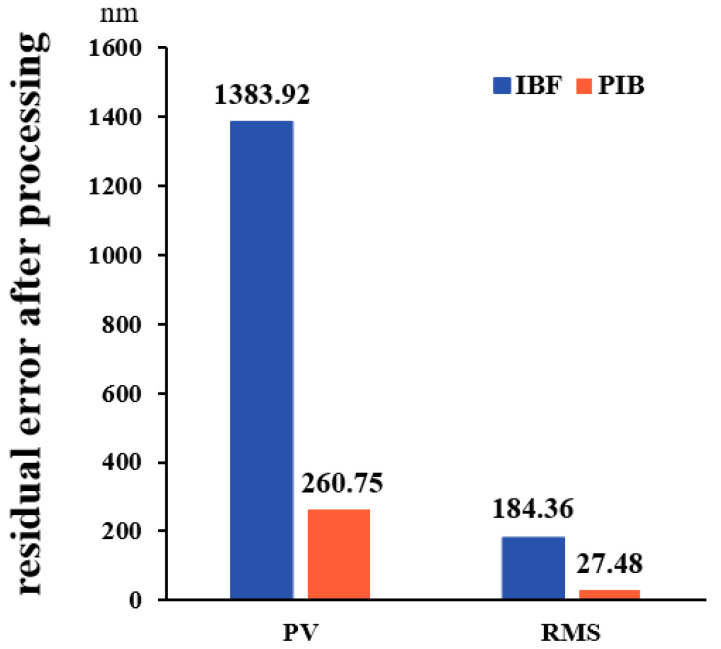
Comparison of the residual convergence after processing.

**Figure 6 micromachines-13-01159-f006:**
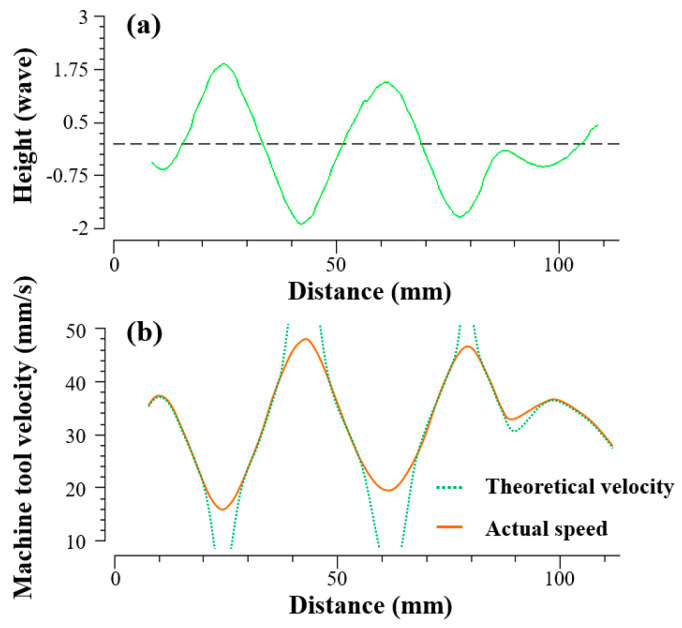
(**a**) Large gradient error to be processed. (**b**) Comparison of the ideal and actual speed of the machine tool.

**Table 1 micromachines-13-01159-t001:** The PIB and IBF processing parameters.

Parameter	Value	Parameter	Value
Ion energy	600 eV	Duty cycle (PIB)	0–60%
Frequency (PIB)	10 Hz	Pulse width (PIB)	0–60 s
Ion Species	Ar+	Sputtering angle	90°

## Data Availability

The data presented in this study are available on request from the corresponding author. The data are not publicly available due to the data also forming part of an ongoing study.

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
