# Peer review of "Figuring Method of High Convergence Ratio for Pulsed Ion Beams Based on Frequency-Domain Parameter Control"

_micromachines, 2022, doi:10.3390/mi13081159_

Round 1

Reviewer 1 Report

The authors have proposed a new mechanism to remove material from 2-D surfaces. This is based on the adjustable duty cycle of the ion beam that emitted energy on the surface. The manuscript novelty is not clear, the written style needs major improvement before it gets published in Micromachines. Below are some concerns that need to be addressed:

- Typo in line #20: " surfaces. And its processing"

- The authors claimed that "This work demonstrates that PIB has better residual convergence in large gradient surface processing." Please elaborate on this in the manuscript to show the visibility of this method compared to the others.  

- Re-write "Previously, Zhou et al.[19] in our research group only validated the stability". 

- Add space before each citation. For example, lines# 34,36, 38....

- "The PIB modification solves this problem in principle". It does not read well. 

- Could you please provide any results that validate this sentence "The amount of removal is linearly related to the duty cycle"?

- Lines #107-109 do not read well. This is misleading here. 

- The reference list does not cover a wide area of this work. Some of them are self-citations. It is okay but you need to report other related works.

- The manuscript addresses a fundamental problem, however, the way it was presented needs a major and intensive revision. 

- The language and writing style can not help the reader to understand and pick the novelty of this work. 

- Please re-write the conclusion as a single paragram instead of a list of points. 

Author Response

Response to Reviewer 1 Comments

Point 1: Typo in line #20: " surfaces. And its processing"

Response 1: The corresponding part has been modified as suggested.

Point 2: The authors claimed that "This work demonstrates that PIB has better residual convergence in large gradient surface processing." Please elaborate on this in the manuscript to show the visibility of this method compared to the others.

Response 2: According to your suggestion, the manuscript has been modified accordingly. We use the index of "convergence ratio" to evaluate the new PIB and the traditional IBF. The advantages of PIB are also described in the revised manuscript.

Point 3: Re-write "Previously, Zhou et al.[19] in our research group only validated the stability".

Response 3: The corresponding part has been modified as suggested.

Point 4: Add space before each citation. For example, lines# 34,36, 38....

Response 4: The corresponding part has been modified as suggested.

Point 5: "The PIB modification solves this problem in principle". It does not read well.

Response 5: This sentence has been revised in the resubmitted manuscript.

Point 6: Could you please provide any results that validate this sentence "The amount of removal is linearly related to the duty cycle"?

Response 6: We have verified this in the early stage, and have not explained it again in consideration of the independence of the manuscript. There is a very detailed description in Section 3.2 of the quoted literature. The quoted source is marked at the corresponding place in the resubmitted manuscript. For your convenience, I will show the results in the attachment.

Point 7: Lines #107-109 do not read well. This is misleading here.

Response 7: This part has been revised in the resubmitted manuscript.

Point 8: The reference list does not cover a wide area of this work. Some of them are self-citations. It is okay but you need to report other related works.

Response 8: In the resubmitted manuscript, the references were updated to be more relevant to the work, and some self-citations were deleted.

Point 9: The manuscript addresses a fundamental problem, however, the way it was presented needs a major and intensive revision

Response 9: We have revised the performance of this work to make it easier to understand.

Point 10: The language and writing style can not help the reader to understand and pick the novelty of this work.

Response 10: We have improved the expression of the result, making its novelty easier to understand. At the same time, we have polished the text of the whole manuscript to make it more readable.

Point 11: Please re-write the conclusion as a single paragram instead of a list of points.

Response 11: The corresponding part has been modified as suggested.

Reviewer 2 Report

The manuscript from Lingbo Xie et al. entitled “Figuring Method of High Convergence Ratio for Pulsed Ion Beam Based on Frequency Domain Parameter Control” show a comparison between pulsed ion beam and continuous ion beam figuring for the material modification with micro and nano resolution. The presented results and theoretical background showed are interesting and the manuscript is valid for publication. Before to be published, however, the paper should be revisited to improve its readability.

I will suggest the authors to improve the background description, too many aspects are taken for granted and without description, an improvement in this sense will broaden the possible audience that can be reached by such a publication. For example, many acronyms should be described fully first time that are used.

In addition, please improve the explanation of results from the simulations and from the experiment, currently are not clearly explained and the reading of the paper is complicate.

Discussion and Results section are well explained and well consistent with the text and results, focalize the previous suggested modification in the previous section  

Author Response

We presented the background of this study in greater detail in the resubmitted manuscript and clarified all abbreviations, making the manuscript more understandable to a wider audience.

Furthermore, in response to your suggestions, we have detailed explained the simulation and testing results so that readers may grasp the novelty of our study. At the same time, we refined the entire manuscript's text to make it more readable.

Thank you for your review, wish everything goes well with your work!

Round 2

Reviewer 1 Report

Dear Authors,

Could you please send me a clear version with highlights only? It is not easy to track the changes in the current version.

Thanks

Reviewer 2 Report

After the corrections made by the authors the manuscript now is improved and can be published as it is. 

Round 3

Reviewer 1 Report

Thank your for the updated version. The manuscript deserves to be published in Micromachines.